# Study on the Influence of Mud Properties on the Stability of Excavated Face of Slurry Shield and the Quality of Filter Cake Formation

**Hu Wang [1], Jian Chen [1,2], Hongjun Liu [1], Lei Guo [3,\*], Yao Lu [1], Xiuting Su [1], Yongmao Zhu [1] and Tao Liu [1,4,5,\*]**

[1]   College of Environmental Science and Engineering, Ocean University of China, Qingdao 266100, China; wanghu@ouc.edu.cn (H.W.); Chenjian1018@163.com (J.C.); hongjun@ouc.edu.cn (H.L.); luyao@stu.ouc.edu.cn (Y.L.); sxting0920@163.com (X.S.); zhuyongmao@stu.ouc.edu.cn (Y.Z.)
[2]   China Railway 14 Bureau Group Co., LTD., Jinan 250101, China
[3]   Institute of Marine Science and Technology, Shandong University, Qingdao 266000, China
[4]   Key Laboratory of Shandong Province for Marine Environment and Geological Engineering, Ocean University of China, Qingdao 266100, China
[5]   Laboratory for Marine Geology, Qingdao National Laboratory, Qingdao 266000, Shandong, China
\*   Correspondence: rendar_lx@163.com (L.G.); ltmilan@ouc.edu.cn (T.L.); Tel.: +86-0532-6678-1020 (T.L.)

**Abstract:** In the construction of subsea tunnels, the stability and control of the excavation surface are the main concerns of the engineering community. In this paper, the Xiamen Metro Line 2 is used as the study case. The filter cake formation of mud shields is experimentally studied, and the excavation surface is numerically simulated. It is found that the formation of filter cake does not require a large pressure difference, and can be formed under 0.06 MPa. With the increase of pressure, the quality of filter cake is further improved, and a small amount of seawater (volume ratio less than 3%) also has a significant effect on the viscosity of mud. Under different cross-section geological conditions, with the decrease of the support pressure of the excavation face, the vertical displacement and vertical (Y-direction) displacement of the excavation face dome gradually increase, the maximum longitudinal displacement is 9.7 mm, the maximum longitudinal displacement can reach 23.9 mm, and the growth trend is nonlinear. According to different stratum conditions, during the excavation of the tunnel, the plastic area of the excavation face is different.

**Keywords:** shield; filter cake; excavation surface; stability

## 1. Introduction

China has become one of the countries with the largest application of shield technology in the world [1]. In the past 10 years, the number of tunnels constructed by the shield method has increased significantly, especially in the construction of urban subways, river crossings and undersea tunnels [2], such as the Nanjing Yangtze River Tunnel, Qingdao Jiaozhou Bay Subsea Tunnel, Xiamen Rail Transit Line 2 and Cross-sea Section of Line 3 and other projects [3,4]. Line 2 from Xiamen Haicang Avenue Station to Dongdu Road Station is China's first cross-sea subsea tunnel constructed with slurry shield machine. The pumps send the slurry prepared on the ground to the pressure chamber. A dynamic filter cake is formed on the excavation surface when the mud is seeping into the formation. The mud pressure of the pressure chamber is balanced by the filter cake and the earth pressure and water pressure on the excavation surface to maintain the stability of the excavation surface. During the construction of long-distance shield tunnel, it is necessary to open warehouses, repair and replace worn cutter discs and knives [5]. Under the adverse conditions of underground seepage field and

seawater intrusion, the excavation surface often collapses, water gushing accident, mud deterioration and other accidents, when it is bad, it will cause personal safety and even the entire tunnel to be scrapped [6]. Therefore, in the construction of subsea shield tunnels, the stability and control of the excavation face of the shield tunnel into the warehouse has become a major topic of concern in the engineering community.

Some examples of forced shutdown and opening of shield tunnels have been reported in the world. Martin and Bapple reported an example of the opening of the fourth tunnel of the Elbe in Germany at the bottom of the river. Heijboer [7] introduced the example of the active opening of the Westerschelde tunnel in the Netherlands [8]. Although there are many examples of successful cabin opening in China and abroad [9–12], most of them are organized by engineers and technicians based on experience, and they are biased towards the construction process, and there is still a lack of theoretical research on the stability guarantee of the excavation face during cabin opening.

The invention of the shield machine started with maintaining the stability of the excavation face. The research on the theory and method of evaluating the stability of excavation face is continuously deepened. The current research methods mainly include theoretical analysis and research, indoor physical test and computer numerical simulation. The theories used to analyze the active instability of excavation faces include limit analysis method and limit equilibrium method. The limit analysis method includes the upper limit analysis method and the lower limit analysis method. Davis et al. gave the upper limit solution of four failure modes under the assumption of the failure surface [13]. Leca used energy dissipation theory to give upper bound solutions for three failure modes [14]. These two analysis methods have the same point in that they both assume a failure mode, that is, the form of the failure surface. Lee et al. applied this upper limit solution to the analysis of the seepage force of the excavated face [15]. Davis gave the elastic lower bound solution of the supporting force without considering gravity. Leca applied this solution to friction-type materials. In addition, many scholars have verified and optimized the classic theory of active instability of shield excavation face through a large number of scale model tests and centrifugal model tests. Takano et al. introduced CT technology to a shield excavation face model test [16]; Kirsch first applied PIV technology to the excavation face stability model test [17]. Mair used a three-dimensional centrifugal model test to study the stability of the excavation face of a shield tunnel in clay [18].

With the continuous development of computer technology, many scholars have used finite element or boundary element technology to establish two-dimensional or three-dimensional models that simulate tunnel excavation, and have achieved considerable results. Zhu et al. established a two-dimensional finite element calculation model. Materials and segments are simulated to provide a method basis for numerical simulation of shield construction [19]. Ding et al. established a two-dimensional finite element model to analyze the influence of in-situ stress on the strength and deformation of the surrounding rock of the tunnel. [20], Yang et al. established a finite difference model to simulate tunnel digging, and provided ideas for the numerical simulation of the stability of the excavation face [21]. In addition, Zhang et al. [22], Jiang et al. [23], Li et al. [24], Zhang et al. [25] and others established three-dimensional numerical calculation models to simulate the excavation of shield tunnels, and studied the ground deformation caused by shield construction. These studies provide ideas for numerical simulation of excavation stability. During the excavation process of the slurry shield, the pressurized mud forms an air-tight filter cake on the excavation face [26], which converts the mud pressure to resist water and earth pressure in front of the excavation face, ensuring the stability of the excavation face. This is especially the case when entering warehouses [27]. The conversion of mud pressure is related to many factors, such as mud properties, mud pressure, formation characteristics, and the form and quality of filter cake formation, etc. [28–30] However, it is difficult to observe the formation process of filter cake in actual engineering. Therefore, it is possible to simulate the permeation process of pressurized mud in the formation by means of laboratory tests, analyze whether the filter cake can form and form regularity, and change the test conditions to study the influence factors of filter cake formation. Watanabe et al. studied the effect of mud density on mud loss in

experiments on the formation of mud films in high-permeability formations and found that the sand content of mud plays an important role in reducing mud loss [31]. Min et al. conducted filter cake formation tests of different muds in formations with a permeability coefficient of $10^{-4}$ m/s, and it was clear that the clay content in the mud has an important effect on the filter cake quality [32].

The above studies are based on ordinary slurry shields as a research object, and the lack of studies on the excavation face of subsea tunnel shields is insufficient to fully explain the mechanism of filter cake formation. In this paper, the Xiamen Metro Line 2 is used as the study case, and the filter cake formation of the slurry shield is experimentally studied, and the numerical simulation of the excavation face is used to further supplement and enrich the stability of the excavation face and the law of filter cake formation.

## 2. Overview of Shield Project

The Xiamen Metro sea-crossing tunnel crosses the sea for 2760 m. During construction, it passes through highly permeable strata such as sand layers. Among them, the vertical permeability coefficient in the residual sandy clay soil layer is up to $6.09 \times 10^{-4}$ m/s. The vertical permeability coefficient in metamorphic sandstone is up to $3.58 \times 10^{-4}$ m/s, and the vertical permeability coefficient in granite formations is up to $0.73 \times 10^{-4}$ m/s.

The area of construction works is shown in Figure 1. The shield section adopts two sets of φ6700 mm composite slurry shields. The overall line is a "V", first descending to the lowest point and then going up. The maximum slope of downhill is 2.8%, and the maximum slope of uphill is 2.9%. The thickness of the covering soil ranges from 8.7 m~65.7 m, the maximum distance from the tide level to the tunnel is about 55 m, and the maximum soil and water pressure is about 6 bar.

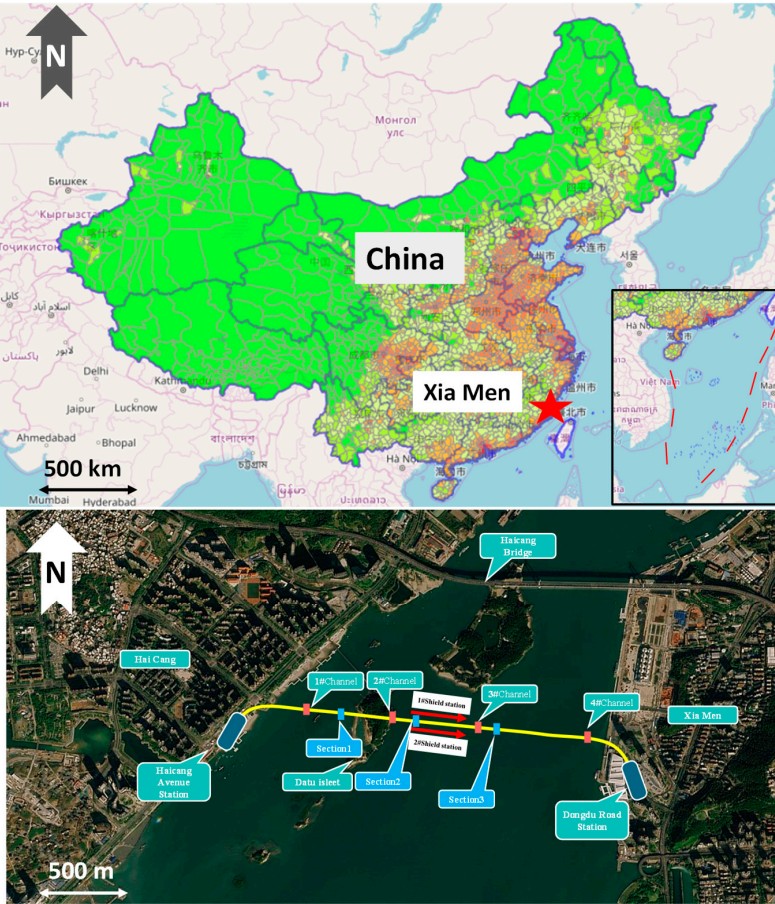

**Figure 1.** Area of construction works.

### 3. Experimental Study on Filter Cake Formation Mechanism

*3.1. Test Materials and Procedures*

The density, viscosity, clay content, and particle size of the slurry have important effects on filter cake quality [33,34]. The infiltration process of the pressurized mud in the formation was simulated by means of laboratory tests, the formation of the filter cake were analyzed, and the test conditions were changed to study the influencing factors of filter cake formation. According to the stratum with high permeability of Xiamen Metro Line 2, different muds were prepared to study the morphological characteristics and formation rules of filter cake formation.

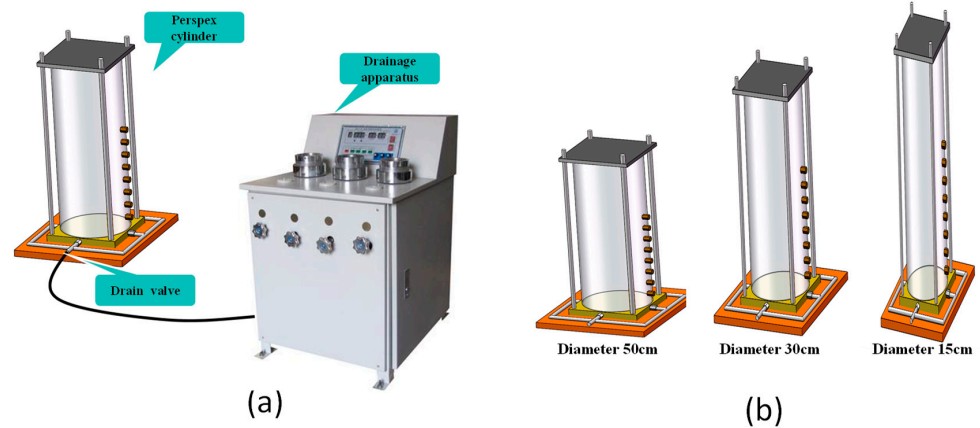

**Figure 2.** Filter cake air Penetration test system. (**a**) drainage apparatus and valve, (**b**) different measuring instruments adopted for the test.

The testing system is mainly divided into drainage apparatus, mud infiltration devices and measuring instruments. The filter cake air Penetration test system is shown in Figure 2. The pressure device can be used to apply pressure to the mud and formation pore water pressure. The mud infiltration device enables the pressure mud to form a filter cake in the simulated formation. There are three specifications for different diameters, namely 15 cm, 30 cm and 50 cm. The data of filtration volume and formation pore water pressure are tested by measuring instruments.

In the experiment, ordinary quartz sand after screening is used to prepare soil sample. Considering the effect of different permeability formations on the formation of filter cake, three types of formation were prepared in the test. The formation parameters are shown in the Table 1. The formation permeability was measured using constant head permeability test. The constant head permeability test is to use the constant head permeability device to measure the seepage flow and the head height of different points, so as to calculate the seepage velocity and hydraulic gradient, so as to calculate the permeability coefficient.

Mud preparation materials include water, granular materials, and additives. Granular materials are mostly clay, bentonite, stone powder, silt, and fine sand, and additives are mostly chemical reagents. See Table 2 for formation test mud parameters.

During the test, referring to the engineering situation of Xiamen Metro Line 2, the maximum loading pressure was set to 0.5 MPa, and adjustments were made according to the test conditions during the test. In this process, a reasonable pressure loading step needs to be set, and the pressure of each stage is set to 0.02 MPa in the initial loading stage, and the loading pressure of each stage is adjusted according to the change of the filtered water amount in the later stage. The loading time of each stage of the mud is controlled within 10~20 s, and the amount of filtered water during filter cake formation is collected at the same time.

**Table 1.** Mass percentage of G1, G2 and G3 stratum in different particle size ranges.

| Stratum | | G1 (%) | G2 (%) | G3 (%) |
|---|---|---|---|---|
| Mass percentage in different particle size ranges | 1.6 mm~2.36 mm | - | - | 35–45 |
| | 1.43 mm~1.6 mm | - | - | 15–20 |
| | 1.18 mm~1.43 mm | - | - | 15–25 |
| | 1 mm~1.18 mm | - | - | 10–15 |
| | 0.8 mm~1 mm | - | 15–25 | - |
| | 0.6 mm~0.8 mm | - | 40–50 | - |
| | 0.5 mm~0.6 mm | - | 30–35 | - |
| | 0.4 mm~0.5 mm | 30–35 | - | - |
| | 0.3 mm~0.4 mm | 25–30 | - | - |
| | 0.25 mm~0.3 mm | 25–30 | - | - |
| | <0.25 | 2.5–5 | 2.5–7.5 | 2.5–5 |
| Permeability coefficient (m/s) | | $3.2 \times 10^{-4}$ | $3.3 \times 10^{-3}$ | $1.93 \times 10^{-2}$ |

**Table 2.** Stratum parameter.

| Stratum | Mud Number | Bentonite Slurry | | Clay | Sand | | CMC (mL/1000 g) | Density (g/cm³) | Viscosity (Pa·s) |
|---|---|---|---|---|---|---|---|---|---|
| | | Content (%) | Particle Size (mm) | Content (%) | Particle Size (mm) | Content (%) | | | |
| G1 | 1 | 95 | <0.25 | 5 | - | - | 10 | 1.07 | $1.15 \times 10^{-2}$ |
| | 2 | 97.5 | <0.25 | 2.5 | - | - | - | 1.06 | $1.06 \times 10^{-2}$ |
| | 3 | 95 | <0.25 | 5 | - | - | - | 1.07 | $1.12 \times 10^{-2}$ |
| | 4 | 97.5 | <0.25 | 2.5 | - | - | 10 | 1.06 | $1.15 \times 10^{-2}$ |
| G2 | 1 | 94.5 | <0.25 | 2.5 | 0.3~0.25 | 3 | 7 | 1.07 | $1.26 \times 10^{-2}$ |
| | 2 | 91.5 | <0.25 | 2.5 | 0.3~0.25 | 6 | 7 | 1.08 | $1.36 \times 10^{-2}$ |
| | 3 | 88.5 | <0.25 | 2.5 | 0.3~0.25 | 9 | 7 | 1.11 | $1.43 \times 10^{-2}$ |
| | 4 | 89 | <0.25 | 5 | 0.3~0.25 | 6 | 7 | 1.1 | $1.38 \times 10^{-2}$ |
| | 5 | 92.5 | <0.25 | 7.5 | - | - | 7 | 1.09 | $1.23 \times 10^{-2}$ |
| G3 | 1 | 91.5 | <0.25 | 2.5 | 0.4~0.3 | 6 | 30 | 1.1 | $1.57 \times 10^{-2}$ |
| | 2 | 91.5 | <0.25 | 2.5 | 0.6~0.5 | 6 | 30 | 1.1 | $1.53 \times 10^{-2}$ |
| | 3 | 85 | <0.25 | 5 | 0.8~0.6 | 10 | 20 | 1.12 | $1.75 \times 10^{-2}$ |

Note: The CMC is sodium carboxymethyl cellulose.

*3.2. Filter Cake Formation State*

3.2.1. G1 Formation

The permeability coefficient of G1 formation is $3.2 \times 10^{-4}$ m/s, which is a highly water-permeable formation. The pressure loading of G1 formation mud is divided into 21 levels of loading, and the loading pressure of each stage increases by 0.02 MPa, and the maximum loading pressure is 0.42 MPa. During the loading process, the filtered water quantity is read every 5 s.

The formation and morphology of filter cake are shown in Figure 3. The mud pressure is greater than the pore water pressure in the formation, and the water and fine components in the mud will penetrate into the formation through the pores of the formation. The mud particles fill the pores of the formation, making the pores of the formation smaller and the permeability coefficient smaller. Infiltration of water increases pore water pressure in the formation. Mud particles gather on the surface of the formation, forming a filter cake of mud skin type. A small amount of mud particles penetrate into the formation, blocking the pores of the formation.

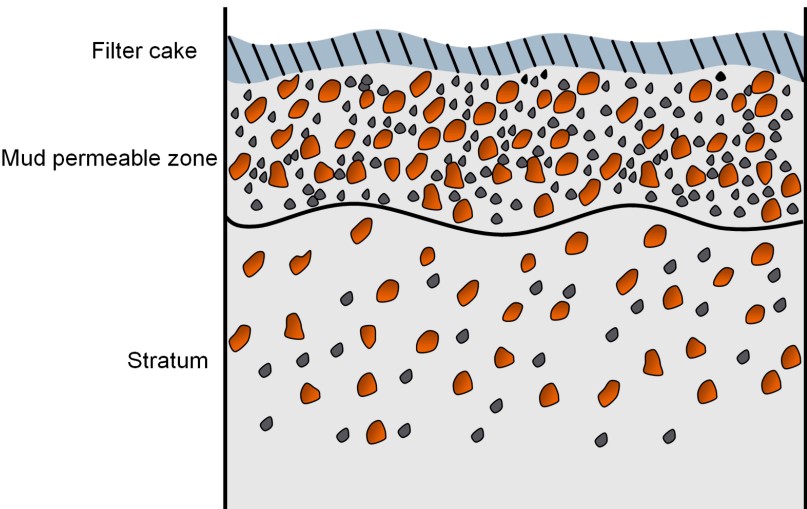

**Figure 3.** Morphology of filter cake (not in scale).

The four types of mud form the "mud skin + permeable zone" filter cake in the G1 formation. A layer of dense mud with a thickness of about 1 mm is formed on the surface of the formation. Under the same clay content, such as 1# mud and 3# mud, 4# mud and 2# mud, the distance of the mud penetrating into the formation becomes larger as the viscosity decreases. Increasing the content of clay can reduce the distance that mud penetrates into the formation, such as 1# mud and 4# mud, 2# mud and 3# mud.

The formation of the filter cake has been completed in the initial stage of loading. As the loading progresses, the quality of the filter cake continues to improve. Figure 4 shows the change in the amount of mud water filtration with time and loading steps, which acts at a smaller initial loading pressure (0.02 MPa). Then, the amount of filtered water increased sharply, and then the growth rate of the filtered water volume slowed down significantly. The entire process was completed within 30 s, and the slowed growth of the filtered water volume indicated the formation of a filter cake. In the subsequent loading process, the amount of filtered water increases rapidly after each step of loading is completed, and the growth rate is much smaller than the initial loading stage, and then the growth of filtered water slows down. After the loading pressure is greater than 0.04 MPa, if the effect of each stage of loading on the increase in the amount of filtered water is ignored, the increase in the amount of filtered water changes linearly, indicating that the quality of the filter cake continues to improve after each stage of loading.

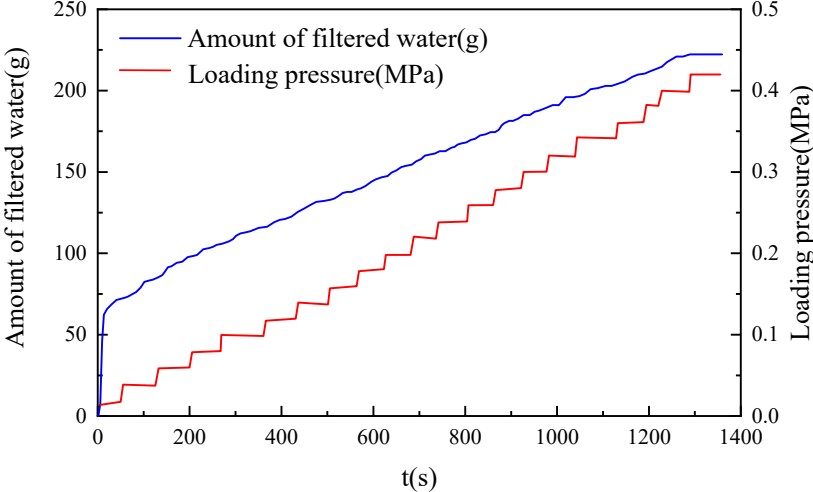

**Figure 4.** The amount of filtered water varies with loading and time in stratum G1.

### 3.2.2. G2 Formation

The filter cake formed in the G2 stratum is in the form of "mud skin + permeable zone" (see Figure 5). Figure 6 shows the changes in the amount of filtered water with the loading process in stratum G2. Among them, the 5# mud failed to form a filter cake. With the increase of clay content and sand content, the thickness of filter cake gradually becomes thicker, the maximum thickness is 3 mm, but the variation is within 2 mm. Only a small amount of sand exists when the filter cake is dissolved in water, indicating that the filter cake is mainly composed of clay particles. From Table 3, at the same clay content, when the sand content increases from 3% to 9%, the mud invasion distance decreases from 5.5 cm to 2.8 cm; when the sand content is 6%, the clay increases from 2.5% to 5%, and the mud invasion distance decreases by 0.4 mm (see Table 3).

**Table 3.** Mud invasion distance (thickness of permeable zone).

| Mud Number | 1 | 2 | 3 | 4 | 5 |
|---|---|---|---|---|---|
| Mud invasion distance (length of permeable zone) (cm) | 5.5 | 3.7 | 2.8 | 3.3 | – |

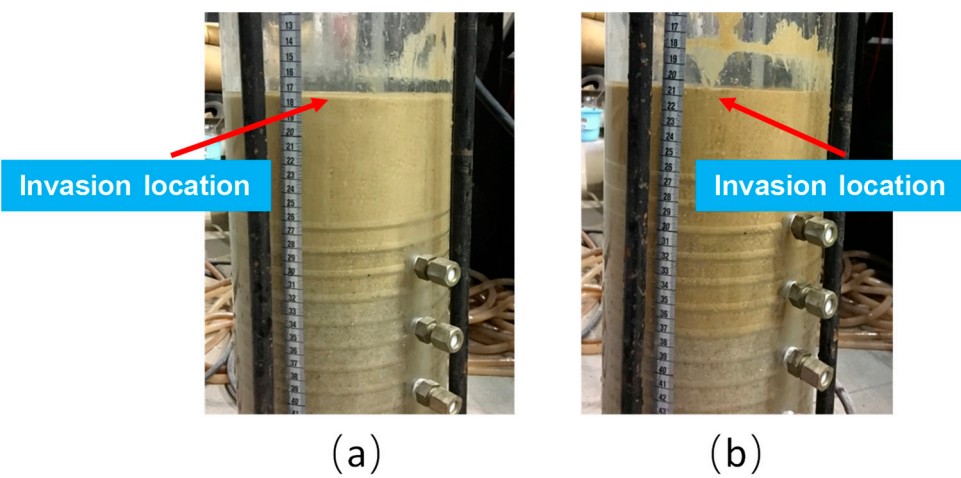

**Figure 5.** Mud intrusion into the formation in stratum G2. (**a**) 3# mud Intrusion Formation; (**b**) 5# mud Intrusion Formation.

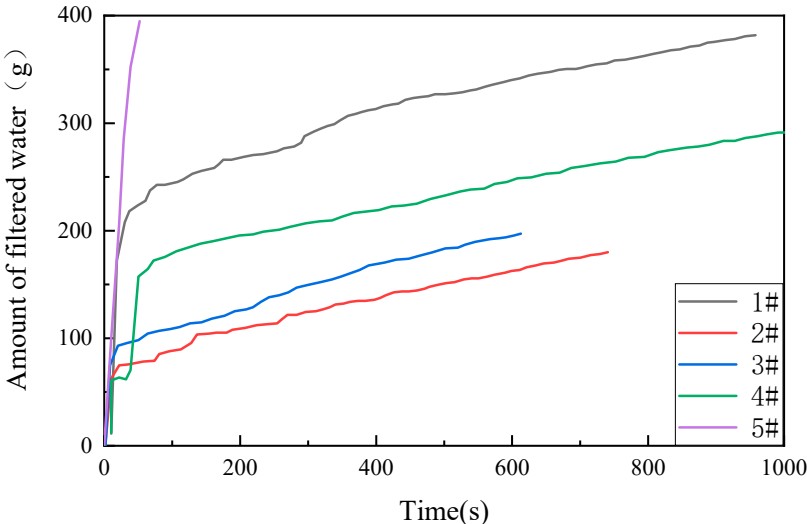

**Figure 6.** Changes in the amount of filtered water with the loading process in stratum G2.

It can be seen from the three stages of the change in the amount of water filtration that the filter cake is basically formed at the end of the first stage. At this stage, the filter cake mainly forms a permeable zone, and the formation still has a certain permeability. In the second stage, due to the pores of the formation are small due to the filling of mud particles, and the small clay particles in the mud gradually accumulate on the surface of the formation to form mud crusts. Because the mud skin is relatively dense, the permeability of the formation is further reduced. In the third stage, the permeability of the formation has become very small. The loading pressure at each stage is relatively high, mud particles continue to accumulate, and the mud skin is compressed and consolidated under pressure, resulting in the permeability of the formation becoming very small.

Because the permeability of the G3 stratum is too large and the permeability coefficient is 1.9 cm/s, the three muds prepared all penetrated into the stratum during initial loading (0.02 MPa), and none of them formed a filter cake in the stratum (see Figure 7).

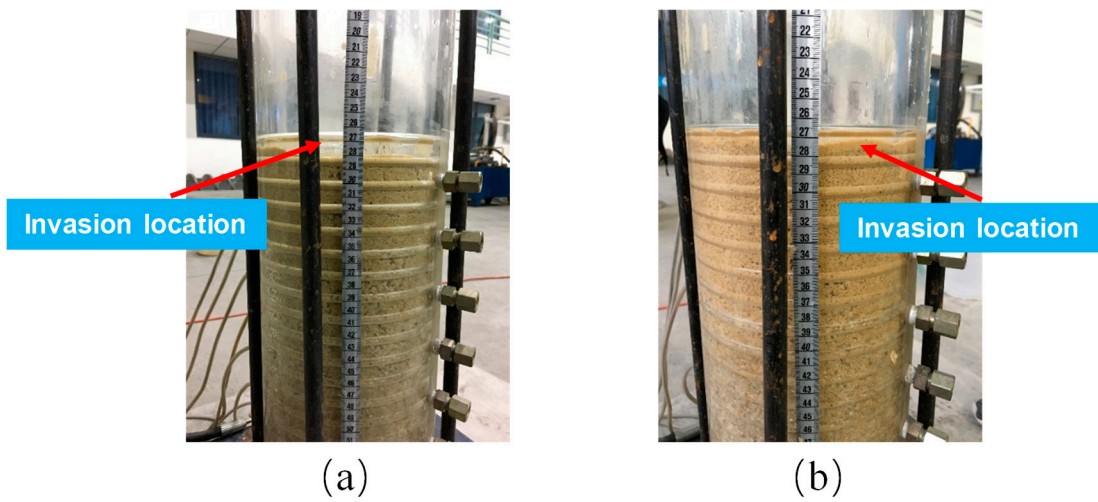

**Figure 7.** Mud invasion state in G3 stratum. (**a**) 1# Mud intrusion formation; (**b**) 3# Mud intrusion formation.

### 3.3. Experimental Study on the Effects of Seawater Erosion on Mud Properties

Because the project is located on the sea floor, and most of the surrounding rocks are fragmented and affected by seawater, the groundwater contains a large amount of $Ca^{2+}$, $Mg^{2+}$, and $Na^+$ ions. After the mud and groundwater are mixed, the stability of the mud is affected. When these positively charged ions change, they change from a suspended state to an agglomerated state, which reduces the particles involved in the formation of the filter cake, reduces the viscosity of the mud, and reduces the ability of the mud to form a filter cake [35,36]. When the slurry shield works, after the salt is mixed with the slurry, the slurry penetrates into the stratum to form a filter cake. A thorough understanding of the mixing characteristics of mud and seawater and the characteristics of filter cake formed by salt cement slurry is the key to mud shield construction.

The test mud was 1# mud of G1. 150 mL of seawater was added to 1500 mL of mud one by one, and the amount of each addition was 15 mL. The test was performed 10 times. In order to eliminate the influence of water in the seawater on the viscosity of the mud, a control group experiment was designed in which the viscosity of the mud changed when tap water was added.

## 4. Numerical Simulation of Excavation Surface Stability

### 4.1. Calculation Model

Under different stratum conditions, the degree of secondary disturbance caused by shield tunnel construction is not the same, and the stability of the tunnel face after excavation is also different. Under complex engineering conditions, it is not enough to rely on simple engineering analogies. It is

necessary to use numerical calculation methods to determine the optimal excavation and support process, and to analyze the dynamic behavior of the tunnel. It is necessary to judge the stability of the caverns [37–40]. In order to evaluate and analyze the stability of the excavation face of shield construction in this project, unfavorable sections with poor geological conditions are selected according to the geological map of the project interval. The FLAC$^{3D}$ finite difference software is used to simulate the construction process of a shield tunnel, and the stability of the shield excavation face under different stratum conditions is analyzed [41].

The numerical model of shield tunneling is established on the platform of FLAC$^{3D}$. In order to minimize the "boundary effect" in the calculation, combined with the excavation diameter of the shield, the overall size of the model is 90 m (*X* axis) × 60 m (*Y* axis) × 55.36 m (*Z* axis); the inner diameter of the segment is 6.0 m, the outer diameter is 6.7 m, the thickness is 0.35 m, and the segment length is 1.2 m. Mohr-Coulomb constitutive model can effectively simulate the mechanical behavior of the formation when it enters the plastic stage; elastic constitutive model is used for segment lining structure. In order to simplify the calculation, different buried depth is simulated by applying different loads on the model. See Table 4 for the calculation parameters of related strata involved in the calculation. It mainly includes physical property test, compression-consolidation test and triaxial shear test of soil. Meanwhile, the mechanical parameters of shield segment lining during model excavation are shown in Table 5.

**Table 4.** The calculation parameters of related strata involved in the calculation.

| Section | Strata Situation | Density (kg/m$^3$) | Elastic Modulus (MPa) | Poisson's Ratio | Cohesion (MPa) | Internal Friction (°) | $K_0$ (coefficient of Horizontal Earth Pressure) |
|---|---|---|---|---|---|---|---|
| 1 | plain fill | 1870 | 7.68 | 0.46 | 20 | 15 | 0.45 |
| | silt | 1620 | 7.68 | 0.35 | 10 | 15 | 0.56 |
| | silty clay | 1960 | 30 | 0.25 | 10 | 4 | 0.54 |
| | residual sandy clay | 1840 | 50 | 0.33 | 22 | 20 | 0.52 |
| | fully weathered granite | 1870 | 80 | 0.3 | 25 | 25 | 0.45 |
| | scattered strongly weathered granite | 1920 | 150 | 0.25 | 30 | 27 | 0.40 |
| | Fragmented strong weathered granite | 2210 | 1300 | 0.2 | 40 | 28 | 0.35 |
| 2 | silt | 1620 | 7.68 | 0.35 | 10 | 15 | 0.62 |
| | fully weathered granite | 1870 | 80 | 0.3 | 25 | 25 | 0.45 |
| | scattered strongly weathered granite | 1920 | 150 | 0.25 | 30 | 27 | 0.42 |
| | moderately weathered granite | 2640 | 2000 | 0.2 | 50 | 30 | 0.35 |
| 3 | silty sandstone | 1780 | 7.68 | 0.38 | 8 | 12 | 0.45 |
| | silt | 1620 | 7.68 | 0.35 | 10 | 15 | 0.60 |
| | silty clay | 1960 | 30 | 0.25 | 30 | 20 | 0.54 |
| | silty clay | 1750 | 7.68 | 0.35 | 12 | 5 | 0.62 |
| | strongly sandstone | 1960 | 140 | 0.25 | 28 | 25 | 0.42 |

**Table 5.** Calculation parameters of shield segment lining.

| Related Parameters | Length (m) | Thickness (m) | Elastic Modulus (MPa) | Poisson's Ratio | Density (kg/m$^3$) |
|---|---|---|---|---|---|
| Shield segment | 1.2 | 0.35 | 34.5 × 10$^3$ | 0.17 | 2460 |

*4.2. Basic Calculation Assumptions*

The assumptions in the calculation are as follows: (1)The ground surface and each soil layer are assumed to be in a uniform horizontal layered distribution; (2) The squeezing and shearing effects of the shield shell itself and the soil body are not considered; (3) The change of soil physical and mechanical parameters during construction is not considered.; (4) The size of the shield tail gap, the degree of

grouting and filling compaction, and the degree and range of disturbance of the soil around the tunnel are not taken into account during shield construction.

### 4.3. Numerical Simulation Calculation Process

This paper mainly focuses on the mechanical behavior and stability of the tunnel excavation surface during shield construction. Therefore, the axial excavation section of the tunnel is set as the target surface in the calculation model to analyze the behavior of the target surface during construction. Therefore, in the process of construction simulation, one-time excavation and support are taken into account before and after the target, i.e., 30 m (25 rings) before the target, 1 excavation step is set at the middle target section, 1.2 m (1 ring segment width) is excavated, and 28.8 m (24 rings) is excavated after the target surface; During each excavation step, shield excavation is simulated and shield segments are assembled.

### 4.4. Calculation of Silo Pressure Simulation Adjustment Strategy

Shield excavation is a progressive process. The focus of this paper is the adjustment strategy of the support pressure of the excavation surface during the construction process. Therefore, the support pressure ratio is used to adjust the support pressure. The calculation formula is as follows:

$$\lambda = \frac{\sigma_s}{\sigma_0} \tag{1}$$

In the above formula, $\sigma_s$ is the support stress of the pressure chamber bulkhead node, and $\sigma_0$ is the lateral static earth pressure of the initial formation. According to Rankine's earth pressure formula, shield position at depth of about 19 m, σh0 at this point is 480 kPa. In the calculation, the support stress ratio of the left and right line construction simulation was adjusted to achieve the simulation of the support pressure of the excavated tunnel face. For different calculation sections, the stability of the shield tunnel excavation face under different support pressure ratios (factors) is analyzed during the calculation; according to the construction process of the shield tunnel, the support pressure ratios (factors) used in the calculation of each section. There are 10 calculation conditions of 0.9, 0.8, 0.7, 0.6, 0.5, 0.4, 0.3, 0.2, 0.1, and 0.05 respectively (Support stress of excavation face/initial formation lateral static earth pressure). The stability of the excavation of the target surface under each calculation condition is compared and analyzed.

### 4.5. Boundary Conditions

Because the shield construction speed is fast and the segments can be assembled in time, the change of groundwater level is not obvious at the first time, so the seepage of water can not be considered during construction, that is, there is no seepage boundary condition. Displacement boundary conditions: lateral displacement is constrained, longitudinal displacement is constrained on the front and back, vertical displacement is constrained on the bottom, and the top of the model is a free surface.

## 5. Discussion

### 5.1. Analysis of Filter Cake Formation Mechanism

In the G1 formation, the higher the clay content and the higher the viscosity in the mud, the slower the growth rate of the filtered water volume and the smaller the final filtered water volume (see Figure 8). It can be seen from Figure 9 that, because the clay content and viscosity of 1# mud is the largest, the growth rate of the water filtration capacity and the final filtration volume are the smallest. As the clay content in the mud decreases and the viscosity decreases, the growth rate of filtered water and the final filtered water volume increase. The change in the amount of filtered water of 2# mud and 3# mud is basically the same, indicating that the problem of insufficient particulate matter can be overcome by adjusting the viscosity of the mud.

It can also be seen from Figure 10 that the formation of the filter cake can be completed at a lower pressure (less than 0.06 MPa). As the pressure increases, the quality of the filter cake further improves, and the amount of filtered water is weakened by the increase in pressure. During the second half of the loading, the amount of water filtered showed a linear change with time, indicating that the filter cake can improve the quality in a short time and offset the effect of increased pressure. At the same time, it is noted that the slopes of the second half of the four muds are basically the same, indicating that at the same pressure In the following, the permeability of the filter cake formed by using the same material to prepare the mud is the same. The properties of the filter cake are related to the mud material, but the ratio of material use is not significant.

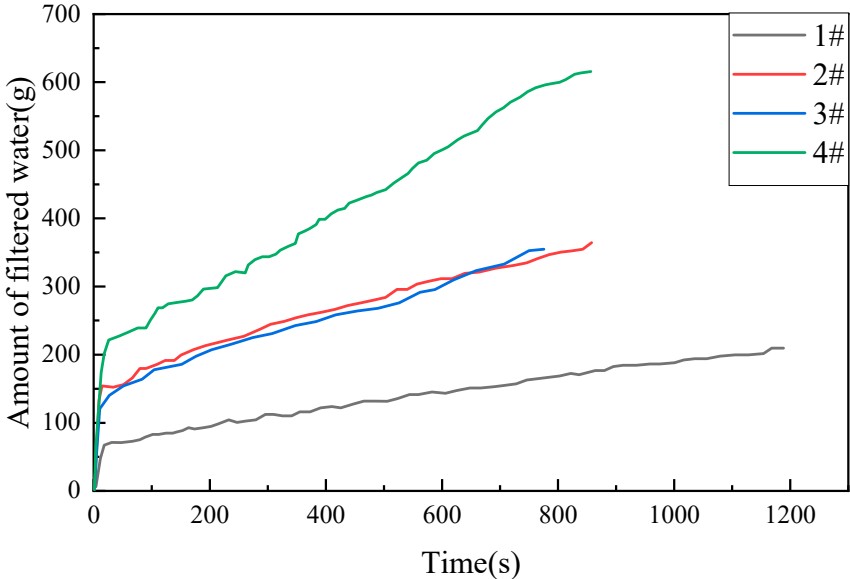

**Figure 8.** Comparison of different mud water filtration capacity in stratum G1.

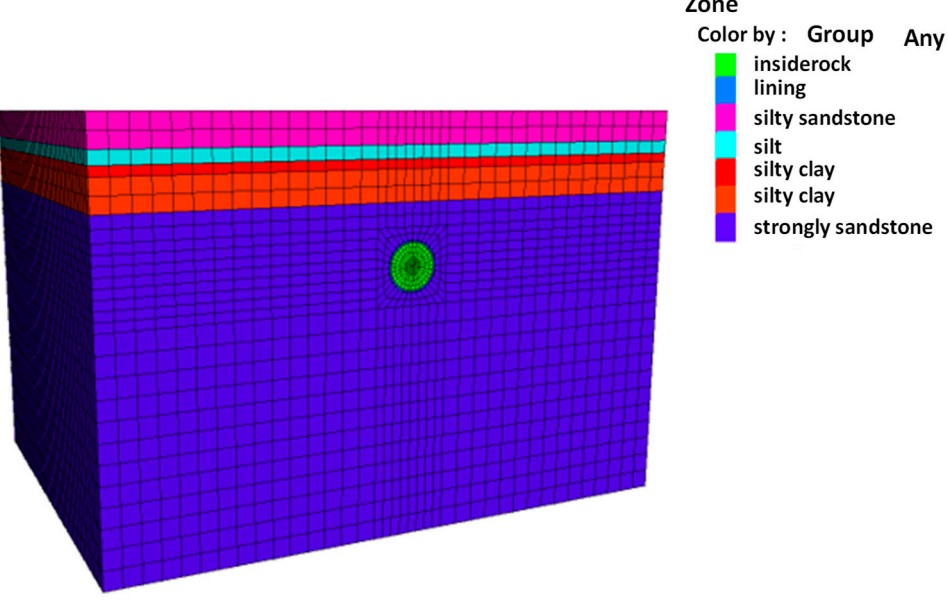

**Figure 9.** The soil profile along Section 3.

No filter cake was formed in G3 formation. There are three main reasons for the analysis: First, although the particle size of sand particles in the mud can block the pores in the formation, but because the particle size distribution of the sand particles in the mud is uniform, it still exists

after blocking the pores in the formation. Mud-permeable channels. Secondly, for highly permeable formations, due to the large porosity of the formation, the mud needs to provide enough particulate matter to block the seepage channels. The content of particulate matter in the mud prepared in the test is not enough to form a filter cake. Lastly, due to the large formation pores, the viscosity of the mud prepared in the test has not yet affected the formation of filter cake.

### 5.2. Effect of Seawater on Mud Viscosity

As can be seen from Figure 10, the addition of seawater reduces the viscosity of the slurry. When the volume ratio of seawater is less than 3%, the viscosity decreases quickly, decreasing by 0.15 Pa·s. After 3%, the viscosity decreases slowly. When the volume ratio of seawater is 10%, the viscosity decreases by 0.15 Pa·s. In the control group test in which tap water was added, when the tap water was added in a volume ratio of less than 7%, the viscosity of the mud decreased substantially linearly, and then the decrease was reduced. Comparing the effect of adding seawater and tap water on the viscosity of the mud, it can be found that after adding more than 7% of the volume, the viscosity of the mud is basically the same, indicating that the influence of ions in the seawater is weakened at this time, and the water content in the mud is mainly affected at this time. It can be seen that even a small amount of seawater (less than 3% by volume) has a significant effect on mud viscosity.

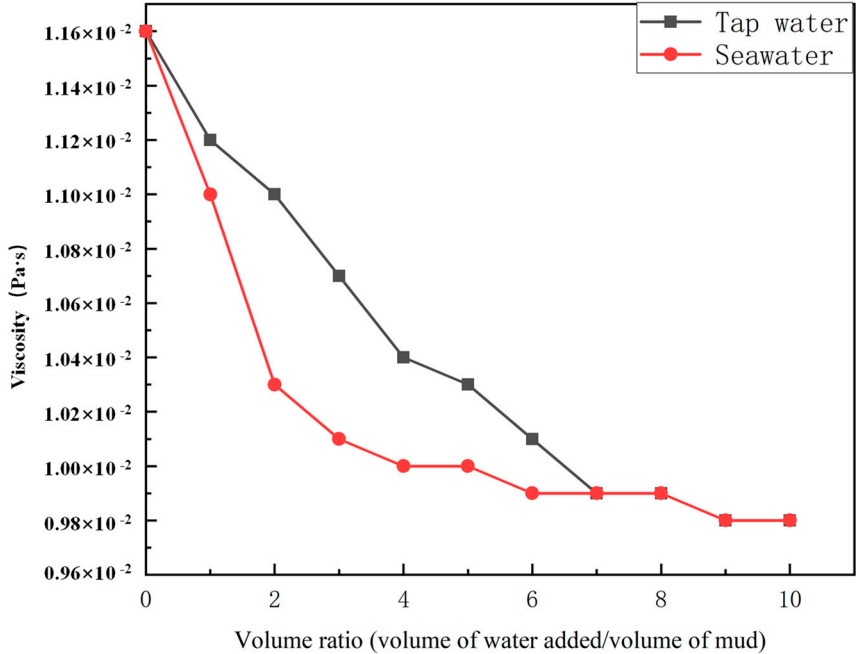

**Figure 10.** Effect of seawater on mud viscosity.

### 5.3. Analysis of Excavation Surface Stability

Under different geological conditions, as the support pressure of the excavation face decreases, the vertical displacement and longitudinal (y-direction) displacement of the excavation face gradually increase, and the growth trend appears non-linear. The maximum vertical displacement of the excavation surface occurs at the vault position, and the longitudinal maximum displacement of the excavation surface occurs at the center position of the tunnel excavation surface (see Figure 11).

According to the data of Section 3 in the Figure 12, during the excavation of shield tunnel, there is a certain settlement at the top of the excavation face and a certain uplift at the bottom of the arch; the vertical displacement of the excavation face is different under different support pressure.

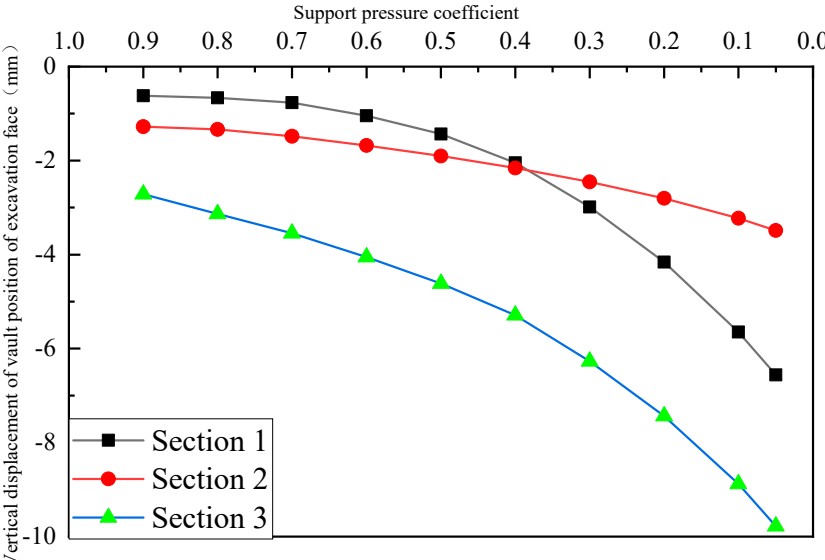

**Figure 11.** Vertical Displacement of Excavation Face under Different Supporting Pressure Coefficients.

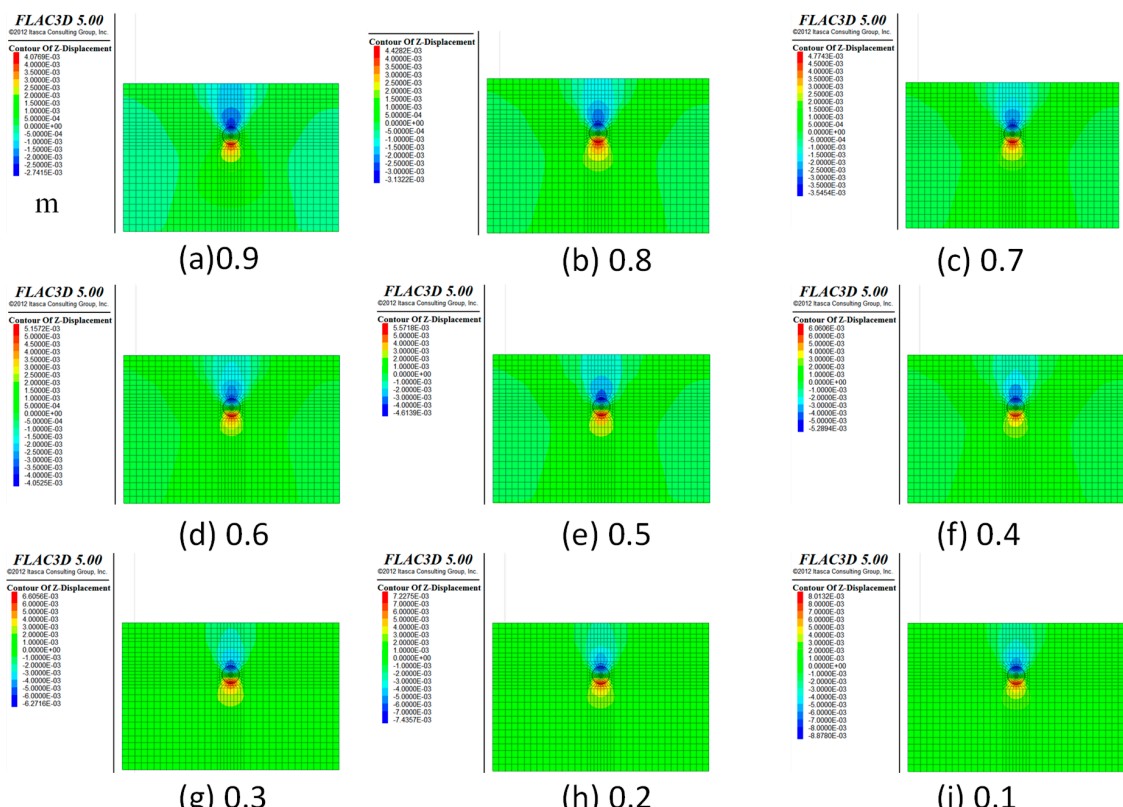

**Figure 12.** Vertical displacement cloud diagram of excavation face under various conditions of Section 3. (The numerical value in the figure represents the support pressure coefficient).

Figure 13 shows variation curve of vault vertical displacement of excavation face under different support pressures. Under different geological conditions, as the support pressure of the excavation face decreases, the plastic area of the excavation face gradually develops; where the support pressure is large, the plastic area of the excavation face is distributed around the tunnel cavity (see Figure 14). Under the condition of small support pressure, the plastic area of the excavation face is relatively large;

according to different stratum conditions, the plastic area of the excavation face has certain differences during the tunnel excavation process.

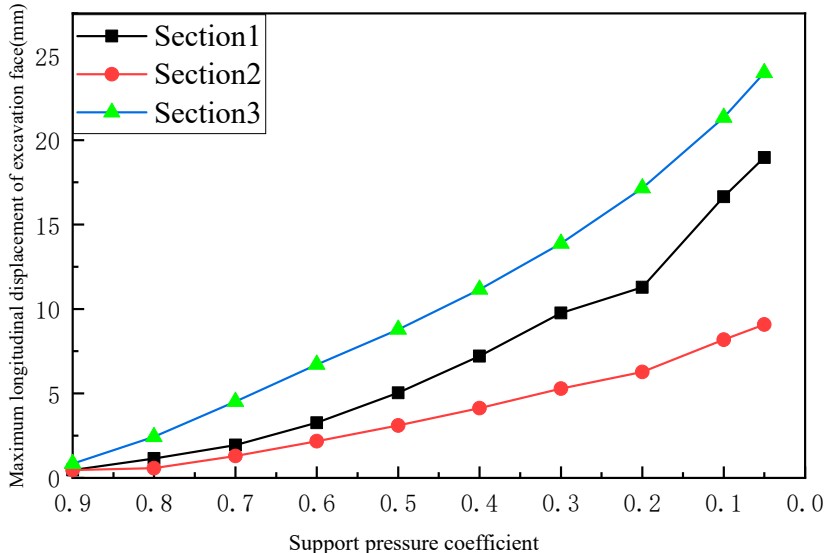

**Figure 13.** Variation curve of vault vertical displacement of excavation face under different support pressures.

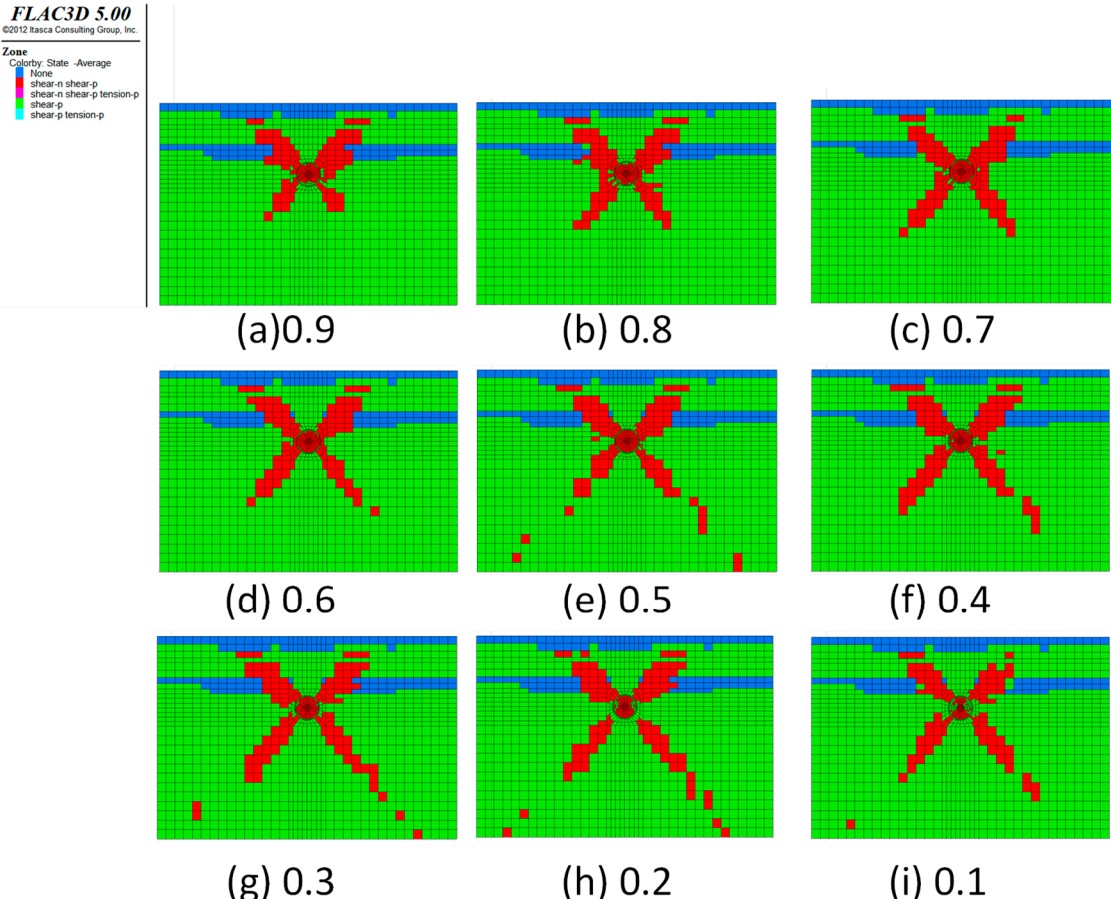

**Figure 14.** Distribution of plastic area in excavation face with different support force coefficients in Section 3. (The numerical value in the figure represents the support pressure coefficient).

## 6. Conclusions

In this paper, the Xiamen Metro Line 2 is used as the engineering study case, the formation of filter cake in slurry shield was studied experimentally. By preparing mud with different properties and carrying out filter cake formation tests in three different strata, the formation patterns and formation laws of filter cakes are tested. The effects of stratum and mud properties on filter cake formation were studied, and numerical simulations of the stability of the excavation face were carried out. The following conclusions can be drawn:

(1) The filter cake has been formed in the initial loading stage, the formation of a filter cake does not require a large pressure difference. As the loading pressure increases, the quality of the filter cake continuously improves;

(2) For strongly permeable formations, filter cakes mainly in the form of "mud skin + permeable zone" are formed, in which the mud skin is thin and the permeable zone is thick;

(3) By increasing the content of clay in the mud, the particle size and content of sand, and the viscosity of the mud, a better-quality filter cake can be formed, reducing the penetration of mud into the formation, and reducing the amount of mud needed to form a filter cake;

(4) Filter cake can be formed only by bentonite base slurry and a small amount of clay in formations with low permeability, but as the permeability of the formation becomes larger and the pores in the formation increase and the pore size increases, the particles in the mud need to be increased. The content of the substance and the viscosity of the mud usually need to ensure a certain sand content in the mud and increase the viscosity of the mud in the project. However, if the permeability of the formation is too large, the gradation of large particles in the mud needs to be considered in order to form a filter cake.

(2) The stratum traversed by the tunnel has good self-stabilizing ability. The stratum that is prone to instability is mainly medium-sand-bearing stratum. However, considering that this project is a subsea shield tunnel, it is affected by high water pressure, fissure development, and shield tunneling. The excavation disturbs the stratum stress field and seepage field, and there is uncertainty in the stability of the excavation face during the opening of the warehouse, which threatens the safety of the construction. How to use and improve the stability of the stratum has become the focus of the project.

**Author Contributions:** Conceptualization, H.W.; methodology, J.C.; software, T.L.; validation, Y.L.; formal analysis, L.G.; investigation, T.L.; resources, Y.Z.; data curation, X.S.; writing—original draft preparation, H.W.; writing—review and editing, H.L.; visualization, H.L.; supervision, L.G.; project administration, J.C.; funding acquisition, T.L. All authors have read and agreed to the published version of the manuscript.

**Funding:** This study was supported by the National Natural Science Foundation of China (Project Nos. 41672272 and 41806075), the Fundamental Research Funds for the Central Universities (201962011), and the foundation for Shandong Province Key Research and Development Plan (Grant No. 201809260098). We also thank Southwest Jiaotong University for providing experimental conditions.

**Acknowledgments:** We appreciate anonymous reviewers who gave comments to revise the paper.

**Conflicts of Interest:** The authors declare no conflict of interest.

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
