# Peer review of "Study on the Influence of Mud Properties on the Stability of Excavated Face of Slurry Shield and the Quality of Filter Cake Formation"

_jmse, doi:10.3390/jmse8040291_

Round 1
Reviewer 1 Report
The paper has been revised and the authors followed the reviewer's answer and suggestions.
Some minor changes:
Line 127: typos error: vale->valve
Line 158: cm/s ->convert o m/s as the rest of the manuscript;
Figure 3 has been added but is not cited in the text. Also, figure 3 needs a size label (approximate) of the various thickness. May be must be added in the caption “not in scale”. Specify what is the “stratum”.
Figure 5 and 7: I suggest to add an arrow or a symbol to put the attention of the reader where is the mud level/intrusion.
I suggest also to change, in table 3, “length” to “thickness”, and also in the text.
Line 246 change “width” to “length”.
Table 5: change “width” to “thickness”
In table 5 there are not all soils indicate in figure 8.
Line 324: there is some issue in the sentence, it is not clear.
Figure 12: the scale legend is very small. I suggest using a less number of labels. The scale MUST be the same for all models, from (a) to (i). This also allows to remove all scale legend and put just one more clear. The picture (b) is not aligned with (a) and (c)
Figure 14: as figure 12, all the color scale can be removed as they are all equals, just put one more clear.
Reviewer 2 Report
The corrections done during the first and second review round have been accepted. Currently, the text is much more readable and the concepts clearer.
However, the paper still presents some minor issues/typos that must be corrected:
LINE 25 - "...the maximum vertical displacement is 9.7mm, the maximum vertical displacement can reach 23.9mm..." I supposed that the first or second "vertical" should be changed in "longitudinal" or "transversal" or "horizontal" or something else;
LINE 36 - "The slurry shield sends the slurry pump prepared on the ground..." should be changed in "The pumps send the slurry prepared on the ground...";
LINE 57 - "methods" should be removed;
LINE 111 - "...soil is 8.7 m ~ 65.7 m,..." should be changed in "...soil ranges from 8.7 m to 65.7 m,...";
LINE 119 - "slurry" should be removed;
LINE 255/257 - In figure Figure 8 the FLAC3D legend reports just Soil_1, Soil_2, etc. Aside Soil_1 should be reported the material type "name", such as "Soil_1 = silty sandstone" and so on for the different strata;
LINE 324 - check "...to form a filter cake. lastly, The viscosity of this..." I supposed a revised form with different punctuation;
LINE 374 - I cannot understand the meaning of this phrase "...experimental research on the formation of mud shields in mud and water shields.". It must be rephrased;
LINE 394 - "formation fissure development" should become "crack formation" or "fissure development"
GENERAL: "background" can be substituted, or at least alternate, with "study case".
Once fixed this list of minor issues, the paper can be accepted as it is.
Reviewer 3 Report
Table 4: For parameters in Table 4, the name of laboratory tests or in situ geotechnical experiments is better to be added to manuscript.
P12 L277: The position and section in which the initial lateral earth pressure is 480 kPa should be added to manuscript.
Fig. 8: Revise the legend to show name of soil layers same as Table 4:section 3. Also, revise caption (Figure8 to Figure 8).
P12 L328: Chage figure 11 to figure 10.
